# Herbal Teas and Drinks: Folk Medicine of the Manoor Valley, Lesser Himalaya, Pakistan

**DOI:** 10.3390/plants8120581

**Published:** 2019-12-07

**Authors:** Inayat Ur Rahman, Aftab Afzal, Zafar Iqbal, Robbie Hart, Elsayed Fathi Abd_Allah, Abeer Hashem, Mashail Fahad Alsayed, Farhana Ijaz, Niaz Ali, Muzammil Shah, Rainer W. Bussmann, Eduardo Soares Calixto

**Affiliations:** 1Department of Botany, Hazara University, Mansehra 21300, KP, Pakistan; drzafar.hu@yahoo.com (Z.I.); fbotany@yahoo.com (F.I.); niaz@hu.edu.pk (N.A.); 2William L. Brown Center, Missouri Botanical Garden, 4344 Shaw Blvd, St. Louis, MO 63110, USA; 3Department of Plant Production, College of Food and Agriculture Science, King Saud University, Riyadh 11451, Saudi Arabia; eabdallah@ksu.edu.sa; 4Botany and Microbiology Department, College of Science, King Saud University, Riyadh 11451, Saudi Arabia; 5Mycology and Plant Disease Survey Department, Plant Pathology Research Institute, Agriculture Research Center, Giza 12619, Egypt; 6Department of Biological Sciences, Faculty of Science, King Abdulaziz University, Jeddah 21589, Saudi Arabia; 7Department of Ethnobotany, Institute of Botany, Ilia State University, 1 Botanical Street, Tbilisi 0105, Georgia; 8Department of Biology, University of Sao Paolo, SP 05315-970, Brazil; 9Department of Biology, University of Missouri, St. Louis, MO 63166, USA

**Keywords:** medicinal plants, herbal teas, traditional knowledge, cultural medicine, Himalayas

## Abstract

In spite of the remarkable achievements in the healthcare sector over recent decades, inequities in accessibility and affordability of these facilities coexist throughout Pakistan. Thus, we aimed to explore and document the cultural knowledge of herbal teas used medicinally by the local community members of Manoor Valley, Pakistan. Field investigations were undertaken during the summer season of 2015–2017, and cultural practices of medicinal plant usage for treating various ailments were gathered through interviews of the local inhabitants. Ethnomedicinal insights of the medicinal plants used in herbal teas were gained with different indexes. Our results revealed 27 plant species, comprising of herbs (70%), shrubs (26%), and trees (4%), which were used for treating 21 diseases. Plants belonged to 18 families: Asteraceae and Lamiaceae were the leading families used for treating diseases. Diarrhea and gas troubles were the most frequent diseases. Based on indexes values, *Cannabis*
*sativa* was the dominant species used. The results revealed that 57% of medicinal uses are new to literature. This ethnomedicinal study is providing the first insights into the traditional medication system of Lesser Himalaya, Pakistan, through ethnomedicinal teas.

## 1. Introduction

Herbal teas are forms of infusions made from a herbal mixture of leaves, seeds, and/or roots of various plants and hot water [1], commonly ingested for their remedial and invigorating properties, including to induce relaxation [2]. Having the capacity to help with stomach related issues, herbal teas often have purging properties and strengthen the immune system. Distinct herbs may possess unique medicinal properties; for instance, herbal teas are generally known for their soothing properties and lower blood pressure [1].

The healthcare systems of many developing countries are based on traditional herbal remedies. About 80% of the human population depends on traditional ethno-remedies of plant origin, and three-fourths of the world population cannot afford modern medicines [3]. Pakistan has a rich history of folk use of plants, with people, especially those living in remote villages, using indigenous plants as medicines, and this knowledge has been passed orally from generation to generation [4,5,6]. Cultural treatments have been used by individuals in Pakistan who have confidence in traditional healers, pastors, hakims, or homeopaths. Health issues like infertility, epilepsy, psychosomatic issues, and numerous different afflictions are often treated with folk medicine. Studies have revealed that most plants have chemical compounds and play a vital role in biological activities [7].

In this context, we observe the importance of studies related to ethnobotany and ethnomedicine for populations in remote regions with low access to health centers. Knowing the methods and use of plants by these populations is a way of understanding the floristic composition of these regions, how these plants can be used to treat diseases, the impacts that the removal and use of these plants can have on the ecosystem, as well as different direct and indirect consequences influenced by these activities. This is the first ever documentation of ethnomedicinal practices using herbal teas for curing various diseases and disorders in the Mansehra region (Pakistan). This study was initiated to explore and document the ethnomedicinal knowledge and local recipes of herbal teas to cure various ailments.

## 2. Results

A total of 55 local participants were interviewed through questionnaires, including 42 men (21–80 years old) and 13 women (21–60 years old) (Table 1).

### 2.1. Floristic Diversity

We found 27 plant species belonging to 18 families: 70% were herbs, 26% shrubs, and 4% trees. These plants were administered by the local population for various health issues. Asteraceae, Lamiaceae, and Fabaceae, were the leading families used in the medication of 21 various diseases with 4 plant species each (14.81%). The remaining 15 families (i.e., Acanthaceae, Apiaceae, Cannabaceae, Chenopodiaceae, Convolvulaceae, Euphorbiaceae, Lythraceae, Malvaceae, Oxalidaceae, Plantaginaceae, Polygonaceae, Portulacaceae, Rutaceae, Scrophulariaceae, and Verbenaceae) were represented by a single species each (Table 2).

### 2.2. Treated Diseases

The local inhabitants and traditional healers were using these 27 medicinal plant species for treating 21 different diseases (Figure 1). Medicinal plants used in the treatment of diarrhea and gas trouble were the most common and treated with five plant species (12.20%) each, followed by diuretic with four species (9.76%) and cough and indigestion with three species (7.32%) each. Moreover, five diseases (i.e., abdominal pain, diabetes, obesity, fever, and kidney stones) were treated with two medicinal plant species (4.88%). Whereas, the remaining 11 diseases: constipation, dysentery, insomnia, liver problems, bad mouth smell, stomachache, throat infection, typhoid fever, vitamin C deficiency, vomiting, and warmness were cured by single medicinal taxa. During interviews, the local informants and especially herbal practitioners were questioned about the safe dosages of the medicinal plants. Locally and traditionally prescribed dosages of various plants were stated in Table 3.

### 2.3. Medicinal Plants Parts Used

The local inhabitants most widely used leaves to cure diseases (36%), followed by the whole plant (22%). Fruits, roots, and seeds were also mentioned by the local informants and traditional practitioners for curing many diseases (14% each) (Figure 2).

### 2.4. Use Value (UVi)

The use value of medicinal plant species ranged from 0.29 to 0.97. Dominant medicinal plants with the highest use value were *Cannabis sativa* having UVi = 0.97, *Mentha longifolia* and *Plantago major* (0.90 each), *Cichorium intybus* and *Oxalis corniculata* (0.86 each), *Bauhinia variegata* (0.82), *Punica granatum* (0.81), and *Medicago sativa* (0.80). The remaining 18 medicinal plant species ranged from 0.78 ≤ to ≤ 0.29 (Table 2). The minimum use value (0.29) was reported for *Xanthium strumarium,* which was used as a diuretic and for the removal of kidney stones.

### 2.5. Relative Frequency Citations (RFCs)

RFCs ranged from 0.20 to 0.65. Most valuable and cited medicinal plant species were *Mentha longifolia* having RFCs = 0.65, *Cannabis sativa* (0.58), *Oxalis corniculata* (0.56), *Punica granatum* and *Cichorium intybus* (0.55 each) (see Table 2). The remaining 22 medicinal plant species were less reported by the local informants, and their RFCs ranged from 0.51 ≤ to ≤ 0.20.

### 2.6. Fidelity Level (%)

Fidelity level ranged from 45% to 100% (Table 4). Medicinal plants with high fidelity level and reported for single medication were *Ajuga integrifolia* and *Taraxacum officinale* for diabetes*, Bauhinia variegata* for obesity*, Dysphania ambrosioides* and *Trifolium repens* for fever*, Cichorium intybus* for typhoid fever, *Indigofera heterantha* and *Portulaca oleracea* as a diuretic*, Malva parviflora* and *Medicago sativa* for gas trouble*, Polygonum plebeium* for cough*, Ricinus communis* for constipation*, Silybum marianum* for liver problems, and *Verbascum thapsus* for diarrhea; each species cited with FL% = 100% for its particular use, respectively. Other medicinal plant species reported with most cited diseases and fidelity values were *Cannabis sativa* (FL = 93.75%) for warmness, *Plantago major* (FL = 92.31%) for diarrhea, and *Mentha longifolia* (FL = 88.89%) for gas trouble. The remaining 10 medicinal plant species reported with a less FL ranged from 45% to %74.19 (Table 4). 

## 3. Discussion

The traditional medicinal usage of plants for curing human ailments is vital to indigenous communities in the northern parts of Pakistan, which is considered valuable local sociocultural heritage [8]. Our study conducted a survey with the local community evaluating the plants, the mode and parts that are used, and for which treatment these plants are indicated. This is the first study related to ethnobotany in this area, which is fundamental for the advancement of folk medicine and for the knowledge and survey of all vegetation used by local communities.

We found a total of 27 plant species belonging to 18 families: 70% herbs, 26% shrubs, and 4% trees, which were administered by the locals for various health issues. As we can see, herbs were the most commonly used type of plant to treat diseases in local medicine. We can link this high use of herbs to the representativeness of this plant growth form in these areas. Many researchers mentioned herbaceous growth form as the dominant plant habit [5,9], which may influence the potential use of these plants to treat diseases. In addition, Asteraceae, Lamiaceae, and Fabaceae were the leading families used in the medication of 21 diseases. Some studies have mentioned Asteraceae as the leading family with the maximum number of plants species for curing diseases [10,11]. For instance, Kumar et al. [12] also reported Asteraceae and Fabaceae as the most significant source of ingredients in traditional folk recipes. 

The local inhabitants and traditional healers use medicinal plant species for treating 21 different diseases. Out of these, diarrhea and gas trouble were the most treated diseases with five plant species each, followed by diuretic with four species, and cough and indigestion with three species each. Stomach and intestinal disorders, as well as cough, are more frequent diseases and generally have a greater knowledge of their treatment by the local community [13]. The local inhabitants most widely used leaves to cure diseases (36%), followed by the whole plant (22%). Leaves are known to have various chemical compounds, which are mainly used as a defense against herbivores [14]. However, these chemical compounds may be beneficial for treating disease, as shown in several studies [5,15]. For instance, Ijaz et al. [15] cited leaves as the most useful phyto-part in medicinal values. Furthermore, Luitel et al. [14] reported that most of the diseases were treated orally. Uses of the plants and their parts by rural inhabitants are common because there is no alternative facility to avail in such remote valleys [6]; as such cultural knowledge is intricately linked to local culture, religion, and history [16]. 

Dominant medicinal plants with the highest use value were *Cannabis sativa* having UVi = 0.97, *Mentha longifolia* and *Plantago major* (0.90 each), *Cichorium intybus* and *Oxalis corniculata* (0.86 each). The maximum use value of reported medicinal plants may indicate their common distribution and/or local practitioner’s expertise, which leads to a priority of choice for ailments [6,17]. In addition, the most valuable and cited medicinal plant species were *Mentha longifolia* having RFCs = 0.65, *Cannabis sativa* (0.58), and *Oxalis corniculata* (0.56). The maximum RFCs reveals that the reported plants are much familiar to the number of local informants and herbal practitioners [17]. Medicinal plants with high UVi and RFCs may indicate high abundance at the region, presence of important active compounds for the treatment of diseases, or even high concentrations of certain active compounds. In this context, these plants should be further evaluated phytochemically to analyze and identify active compounds for potential drug discovery [5]. 

Medicinal plants had a high fidelity level and were reported for single or few medications. These plant species have revealed remarkable medicinal significance, and further assessment through phytochemical, biological, and pharmaceutical activities is important. On an experimental basis, many researchers obtained maximum fidelity level values against oral disorders [18]. Nonetheless, minimum FL may not be abundant, and this highlights the importance of securing these plant species for future generations, as threats of gradual depletion to the extinction of cultural knowledge are rising [19].

The literature has revealed that most plants have chemical compounds and play a vital role in biological activities. For example, *Cannabis sativa* contains important compounds such as alkaloids, cannabinoids, cardiac glycosides, flavonoids, resins, steroids, tetrahydrocannabinol, and cannabidiol. This plant also has significant potential in analgesic, anti-emetic, anti-inflammatory, sedative, anti-convulsive, and laxative actions [7]. Also, *Cannabis sativa* seed oil hepatoprotective and toxicity was also assessed in albino rat, and its safe dosage was also reported [20], showing the potential that the compounds of this plant have for the treatment of diseases. Several other plants have a cocktail of chemical compounds that have active potential against various other diseases, which has been reported in several studies [21,22,23,24,25,26,27]. For instance, *Mentha longifolia* contains anthraquinones, flavonoids, glycosides, lignin, phenolic compounds, saponins, tannins, terpenoids [21]; *Plantago major* contains alkaloid, carbohydrates, coumarin, proteins, glycoside, glycosids, tannins, triterpens [22]; *Cichorium intybus* contains alkaloids, blumeatin, coumarins, diosmetin, dihydroquercetin 7-4′-dimethyl ether, esculetin, flavonoids, glycosides, which have a positive potential as for anti-oxidant, anti-tyrosinase, and anti-bacterial activities [23]. *Ricinus communis* has been reported as an important medicinal plant species in numerous studies, such as purgative, lubricant [28], bowels [29], leg sore, piles, edema [30], purgative, anti-flatulent, anti-inflammatory, analgesic, laxative, diarrhea [13], rheumatic joints, swellings, muscular inflammation, and toothache [31]. Other studies have also shown active plant compounds with the disease they are linked [32,33,34,35,36,37,38,39,40] Although, scientific evidence and proper experimentation for safe or recommended doses as well as lethal and harmful toxicities for most of these species are currently unavailable. 

Comparative analyses of data use of plants for medicinal purposes of this study and published data from regional, national, and global levels based on SQ% and RSI% showed 0% to 27.12% and 0% to 6.78% similarity, respectively. After a cross-cultural comparative analysis of the reported taxa, very fewer similarities were found, and this might indicate the importance of geographical variations. The analytical approach of Sorensen’s similarity index determines only the common floral similarity by comparison of a case study with previous documentation but does not address the common plants with similar use(s) (Appendix A). Our comparative results have 27.12% similarity with Qureshi et al. [28] from Abbottabad, Northern Pakistan, 26.42% with Akhtar et al. [29] from Swat, North Pakistan, 24.39% with Ijaz et al. [5] from Abbottabad, Pakistan, 19.47% with Matin et al. [41] from Shogran, Mansehra, Pakistan and 18.35% with Khan et al. [42] from Naran Valley, Pakistan (Appendix A). Moreover, the remaining 16 studies similarities ranged from 0% to 18.37%, and out of these, four studies showed 0% similarity to Siran Valley, Mansehra, Pakistan [43], Deosai Plateau, Gilgit Baltistan, Pakistan [44], Aladaglar, Nigde-Turkey [45]. On the other hand, the highest degree of similarity index of the present study was found with a study conducted in Abbottabad, Pakistan, by Qureshi et al. [28] with RSI = 6.78%. In comparison, we found an RSI value of 0% with 10 previous studies, and out of these seven studies that had common plant species, no single common plant species has been cited for common medicinal use(s), which is why RSI showed 0% similarity [13,23,31,43,44,45,46,47,48,49] (Appendix A). A cross-cultural analysis of the reported taxa assessed through the RSI% index revealed novel findings and new reports of medicinal plant species used for curing various diseases. 

The cited medicinal plant species were cross-checked with 21 different studies of distinct neighboring, regional, national, and global levels for new/novel medicinal usages (Table 5). Of these, some similar reports were found for species like *Bauhinia variegata* for obesity [5], *Cannabis sativa* as a local cold drink (Tandai) against heatstroke [29], and *Justicia adhatoda* for throat infection [5,28,50] and cough [30]. Similarly, *Ricinus communis* was reported as a remedy for constipation by Matin et al. [41]; Abbasi et al. [51]; Jamal et al. [52], *Salvia moorcroftiana* for liver problems [28], *Silybum marianum* for jaundice, hepatitis, cirrhosis, *Verbascum thapsus* for diarrhea [5,28,41,51]^,^ and *Vitex negundo* for gas troubles [5]. *Mentha longifolia* was being used as a gas expeller [42], for stomach problems [53] and abdominal pain [45]. Further, *Mentha royleana* was mentioned for vomiting and diarrhea [29,42], and *Oxalis corniculata* and *Polygonum plebeium* for bad mouth smell and cough [5], respectively. 

*Ajuga integrifolia* has been reported for the first time nationally for its current medicinal use (diabetes). This plant species is widely used medicinally for jaundice, as well as for hypertension [29,41]. Moreover, *Ajuga integrifolia* was documented for malarial fever [41], abdominal pain [51], throat sore, colic, blood purification and epilepsy [29], tonic, dental, hepatic stimulant [49]. Local community members and traditional healers of the study area used *Bauhinia variegata* to overcome obesity. Whereas, the literature survey suggested different uses like fuel wood [46,50], fodder, vegetable, and ornamental [46], as well as medicinal purposes, such as anthelmintic, tonic, astringent, skin diseases, ulcer, and scrofula [50]. Moreover, *Bauhinia variegata* was reported for gastrointestinal problems by Kichu et al. [13], as mentioned in Table 5.

*Cannabis sativa* has been reported for relieving from warmness and insomnia in the present study, differently from other studies, which have reported for bowels, dyspepsia, gonorrhea, tonic [28], sedative [28,41], anodyne, anti-lice [41], appetite, abdominal swelling [51], narcotic [28,31,41], anti-diarrheal, stimulant [31], and stomachache [13]. *Dysphania ambrosioides* has been reported by local community members for the medication of fever. Furthermore, Matin et al. [41] cited *Dysphania ambrosioides* for piles and stomach problems. Traditional practitioners and informants cited *Cichorium intybus* as effective against typhoid fever. *Cichorium intybus* was cited for different medicinal purposes worldwide, Qureshi et al. [28] documented it for bile secretion, digestion, stomachache, tonic, carminative, cordial, obstructed, menstrual disorder, and dropsy and Akhtar et al. [29] reported *Cichorium intybus* for fever. All these previously published documentations clearly specify the novelty of the current medicinal use of the plant species (Table 5).

*Convolvulus arvensis* is an important medicinal plant species, and in the present study, it has been cited for curing stomach problems, i.e., diarrhea and dysentery. It has also been documented for many different diseases on a national as well as global level [28,29,31,41,46]. Other plants were also reported to treat other diseases not previously reported, such as *Indigofera heterantha, Malva parviflora, Mentha longifolia,* and *Oxalis corniculata.* In the case of *O. corniculata,* it was used by the local inhabitants for Vitamin C deficiency and bad mouth smell; no other researcher has ever cited it for the current use in Pakistan. Whereas in the literature, *Oxalis corniculata* was documented for dissimilar and contradictory medicinal uses, such as anti-septic, skin curing, stomach troubles, cleaning rusted vessel [28,41], and digestion [29].

*Plantago major* has been mentioned in the current study for diarrhea and obesity, in which obesity was the novel use cited in the present study throughout the world. Previously, it was reported for constipation, intestinal problems [41], infected hooves [51], bedsores, and candidiasis [29]. The local inhabitants reported *Polygonum plebeium* for cough. After comparison, contradictory results were found, and *Polygonum plebeium* was reported for pneumonia and bowels with Matin et al. [41]. Several studies reported *Portulaca oleracea* for liver, kidney diseases [29]^,^ abdominal pain, diabetes, asthma [5]. *Portulaca oleracea* has been documented in the present study for diuretic problems. The cited medicinal use of *Punica granatum* in the present study is also novel throughout the world (Table 2). This very important and useful medicinal plant species was reported by many researchers for curing various infections/diseases like cough [29,31,46], dysentery [13,23,46,51], fuel wood, tonic, anthelmintic [29,46], cold, flu, nasal decongestant [31], diarrhea [13], skin diseases [23], heart palpitation, and blood pressure [5].

*Ricinus communis* has been reported as an important medicinal plant species in a number of studies, such as purgative, lubricant [28], bowels [29], leg sore, piles, edema [30], among other diseases. *Salvia moorcroftiana* has been found effective for abscesses, dysentery, and colic by Khan et al. [42] and for wound healing by Akhtar et al. [29]. Furthermore, Ahmad and Pieroni [48] cited the white juicy part of its stem as food. In the literature survey, several important diseases were found to be treated by *Taraxacum officinale* like Hepatitis, appetite [52], urination, kidney disorders [42], tonic [42,53], diuretic, laxative, anti-inflammatory, and digestive disorders [53]. Here, *Trachyspermum amii* is documented for the first time across the globe for diuretic and kidney stone removal. Abbasi et al. [51] cited it as an appetite stimulant and galactagogue. *Trifolium repens* is reported for curing fever, and this plant has not been used previously for the current cited problem. Khan et al. [42] reported the same plant species for worm removal (cattle poison). Many researchers cited *Verbascum thapsus* for various disorders like respiratory tract infection, ear infection, dry cough [52], aphrodisiac, skin problems, irritability [42].

*Vitex negundo* has been cited by the locals for indigestion, stomachache and gas troubles whereas other researchers reported it for variety of uncommon medicinal purposes, like for aromatic, tonic, febrifuge, diuretic, anthelmintic, headache, fuel wood [46], chest pain [23], skin problems, child crying, and snake bites [30]. *Xanthium strumarium* has been reported as a diuretic and for kidney stones. In the literature survey, *Xanthium strumarium* has been cited for skin problems [41], malarial fever [5,41], carminative [5]. *Zanthoxylum armatum* has been cited as a medication of abdominal pain and indigestion in the current study. The literature survey revealed *Zanthoxylum armatum* for different medicinal uses, such as fuel wood, digestion, aromatic, condiment, gum diseases [46], vomiting and indigestion [51], carminative, pile [23,46], malarial fever, antipyretic [29], stomachache [29], fever, and cholera [23] (see also Table 5).

## 4. Materials and Methods 

### 4.1. Study Area

Manoor Valley is a remote area of District Mansehra, situated about 50 km to the North of the main Kaghan road at ‘Mahandri’ (Figure 3). The entire area is formed by crosswise ridges of mountains on either side of the Manoor River, which flows in a northeast to southwest direction down the valley emerged from Malika Partbat (“Queen of Mountains”, elevation 5279 m). Approximately 17,000 inhabitants and 400 families are settled in the area. These residents belong to Gujar, Awan, Mughal, and Swati casts. No other tribes are settled there due to geographic isolation and lack of modern facilities. Moreover, these ethnic groups speak Gujri and Hindko. Fuelwood, timberwood, and selling plants used in herbal Unani medicine, and dry fruits, especially from *Juglans regia*, are the main sources of income along with agriculture. The valley preserves huge biodiversity, especially of plants [54,55], but lack of infrastructure and isolation influences the preservation of this area, resulting in a loss of biodiversity. This geographically isolated community is facing the complete lack of vital basic necessities [6]; due to isolation, most of the population was beyond the scope of this study, but an attempt has been made to highlight medicinal plants of this area that are important healthcare treasures for this community.

### 4.2. Field Surveys, Plants Collection, Interviews, and Questionnaire

Frequent field surveys were undertaken during the early, mid, and late summer season of 2015–2017. Plant specimens were collected, serially tagged, and pressed between the blotting papers for drying [10]. During field visits, interviews were conducted with local participants conducted in Hindko and Urdu languages, after obtaining their oral prior informed consent, in order to get traditionally important medicinal plants information and their folk recipes [10,15]. A semi-structured questionnaire was developed and filled during interviews from local informants, traditional healers, shepherds, and forest guards for vernacular names, as well as cultural practices of medicinal plants for treating various diseases [56,57,58]. Plant specimens and photographs were also shown to the local inhabitants to elucidate the traditional medicinal uses of plant species (Appendix A).

### 4.3. Tea and Drink Infusions Preparation

Fresh leaves of a plant were shade dried, grinded, and powdered. Infusions of leaves or root (depends upon the part used) were prepared in 240 mL of water by adding 15 g of plant powder or extract (amount varies for some species) and boiled. Similarly, for the preparation of drinks, the same amount of powder or extract was added to the water. Sugar was also added to make the taste better. A recipe for each medicinal plant species has been given in Table 3.

### 4.4. Herbarium Work and Plant Identification

Collected specimens were poisoned to prevent herbarium pests, following recommended herbarium techniques, mounted on standard herbarium sheets, and preserved [10]. Plant taxonomist Dr. Abdul Majid identified the plant specimens with the help of Flora of Pakistan [59,60,61]. Identified specimens were deposited in the Herbarium, Department of Botany, Hazara University Mansehra, Pakistan. The scientific name of a plant species was cross-checked and updated with The Plant List website.

### 4.5. Data Analysis

Data collected during interviews were analyzed through use-value (UVi), relative frequency citations (RFCs), and fidelity level (FL%) indices. A cross-cultural analysis of the reported taxa was assessed through the Sørensen’s similarity index (SQ%) and Rahman’s similarity index (RSI%) to evaluate novel uses in comparison to the literature.

#### 4.5.1. Use Value Index (UVi) 

Use value index provides a quantitative value regarding the importance of species for their cited usage. It can be calculated by following [62,63].
(1)$$\mathrm{UVi}=\frac{\mathrm{Ui}}{\mathrm{Ni}}$$
where “Ui” represents the number of use reports mentioned by each informant for a particular plant species, and “Ni” refers to the total number of informants interviewed for that given plant species.

#### 4.5.2. Relative Frequency Citations (RFCs)

The relative frequency citation indicates the importance of each plant species based upon the informants’ citations. It can be determined by following [64].
(2)$$\mathrm{RFCs}=\frac{\mathrm{FCs}}{N}$$
where “FCs” represents the importance of a given plant species use in curing any ailment, and “N” refers to the total number of informants in the study. 

#### 4.5.3. Fidelity Level (FL%)

The fidelity level determines the plant species citations for the most preferred cured ailment. The fidelity level (FL%) can be determined by following [6,65].
(3)$$\mathrm{FL}\%=\frac{Ip}{Iu}$$
where “Ip” refers to the number of informants mentioning use(s) of plant species for a particular ailment, and “Iu” represents the total number of informants mentioning the plant species against any disease.

#### 4.5.4. Sorensen’s Similarity Index (QS%)

Sorensen’s similarity index (QS%) was developed by a botanist Thorvald Sorensen and published in 1948. The comparison with previously published data collected from different regions was performed by evaluating percentages of the quoted species and their medicinal uses by applying the Sorensen similarity index formula [5,17,66].
(4)$$\mathrm{QS}=\frac{2c}{a+b}\times 100$$
where, a = number of species unique in an area A, b = number of species unique in an area B, and c = number of species common to areas A and B.

#### 4.5.5. Rahman’s Similarity Index (RSI)

Rahman’s similarity index (RSI) was developed by Inayat Ur Rahman and Farhana Ijaz and published in 2019. RSI is calculated as “by comparison of the present study with the studies previously published from allied, regional, national, and global levels through the percentages of plant species analysed and commonly cited with same cultural medicinal uses” [67]. The formula used is
(5)$$\mathrm{RSI}=\frac{d}{a+b+c-d}$$
where, “a” is the number of species unique in an area A, “b” is the number of species unique in an area B, “c” is the number of common species in both A and B areas, and “d” is the number of common species used for similar ailment in both A and B areas. While a and b ≠ 0 and c and d ≥ 0. To find out the percentage of common uses between two areas, the formula can be written as
(6)$$\mathrm{RSI}=\frac{d}{a+b+c-d}\times 100$$

## 5. Conclusions

The study area is a remote region with no modern medical facilities available to the local communities; thus, they completely rely on their traditional medicinal practices. This is one of the main reasons that helped to maintain their cultural knowledge and integrity within these geographic boundaries. Our survey resulted in 57% of medicinal uses reported being new, the use of 36% of the medicinal plant species was reported for the first time from Pakistan, and 18% of plant species were reported for the first time globally. A cross-cultural analysis of the reported taxa assessed through the SQ% index revealed new medicinal uses for *Ajuga integrifolia, Dysphania ambrosioides, Cichorium intybus, Convolvulus arvensis, Indigofera heterantha, Malva parviflora, Plantago major, Taraxacum officinale, Xanthium strumarium,* and *Zanthoxylum armatum* for the first time from Pakistan, and *Medicago sativa*, *Portulaca oleracea, Punica granatum, Trachyspermum amii,* and *Trifolium repens*. This documentation not only targeted the traditional practices of plants used for human welfare, but also summarized the active chemical compounds, biological assays, safe and toxic dosages, and results of experimental species. Nonetheless, the sample size of informants, location of the area, and plants used for medication through herbal teas posed limitations. It is also worth mentioning that unless documented, orally transmitted traditional knowledge, such as this, is on the verge of extinction following the death of old practitioners. The current study has important implications for addressing emerging concerns like food security, conservation of ecosystems, and human health. 

## Figures and Tables

**Figure 1 plants-08-00581-f001:**
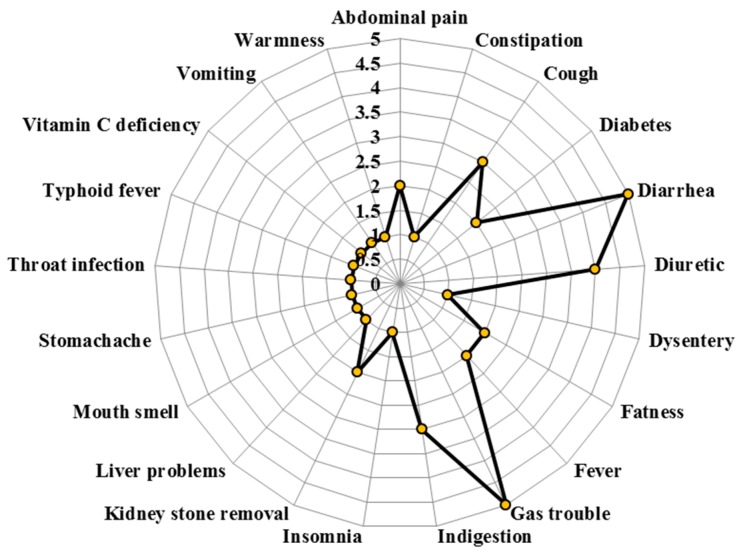
Number of plant species used for the different diseases.

**Figure 2 plants-08-00581-f002:**
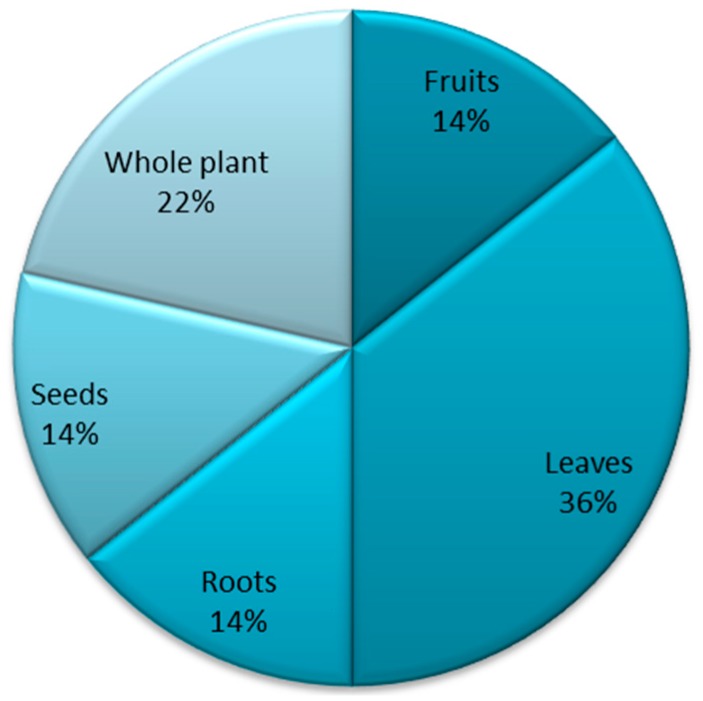
Plants parts used in traditional treatment of different diseases.

**Figure 3 plants-08-00581-f003:**
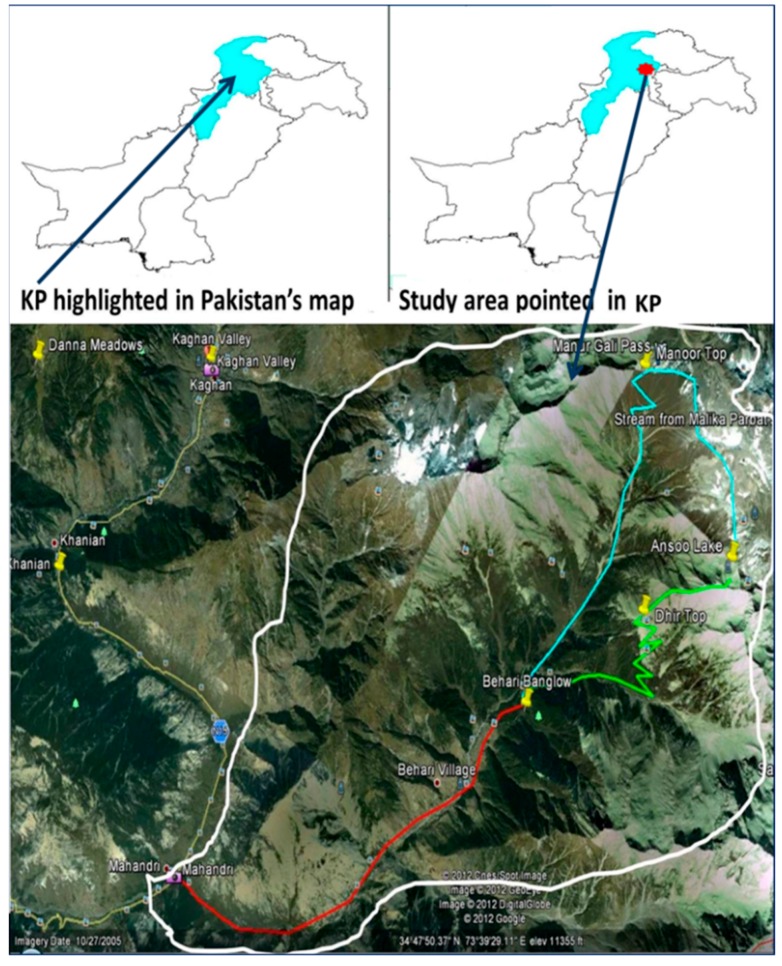
Map of the study area (highlighted in white). (Source: https://www.google.com/earth/).

**Table 1 plants-08-00581-t001:** Number of local inhabitants interviewed during this study.

Gender	Local Inhabitants Age (years)		
	21 to 40	41 to 60	61 to 80
Male	21	14	7
Female	8	5	0
Total	29	19	7

**Table 2 plants-08-00581-t002:** Results obtained during survey of the study area, showing the scientific and vernacular plant name, plant habit (PH: H-Herb, S-Shrub, T-Tree), part used (PU: Wp-Whole plant, R-Root, L-Leaves, Sd-Seeds, Fr-Fruits), medicinal uses (MU), Use Value (UV), Relative Frequency Citations (RFCs), and use reports (URs).

Scientific Name	Vernacular Name (Hindko)	Family	PH	PU	MU	UV_i_	RFC_s_	URs
*Ajuga integrifolia* Buch.-Ham.	Khauri buti	Lamiaceae	H	Wp	Diabetes	0.78	0.45	25
*Bauhinia variegata* L.	Kachnar	Fabaceae	T	R	Obesity	0.82	0.42	23
*Cannabis sativa* L.	Bhang	Cannabaceae	H	L	Warmness, insomnia	0.97	0.58	32
*Cichorium intybus* L.	Hand	Asteraceae	H	R	Typhoid fever	0.86	0.55	30
*Convolvulus arvensis* L.	Laili	Convolvulaceae	H	Wp	Diarrhea, dysentery	0.63	0.35	19
*Dysphaniaambrosioides* (L.) Mosyakin & Clemants	Baljawain	Chenopodiaceae	H	Sd	Fever	0.46	0.22	12
*Indigofera heterantha* Brandis	Kainthi	Fabaceae	S	R	Diuretic	0.68	0.47	26
*Justicia adhatoda* L.	Baiker	Acanthaceae	S	L, R	Throat infection, cough	0.58	0.33	18
*Malva parviflora* L.	Sounchal	Malvaceae	H	L	Gas trouble	0.54	0.27	15
*Medicago sativa* L.	Sengi	Fabaceae	H	L	Gas trouble	0.80	0.51	28
*Mentha longifolia* (L.) L.	Jangli podina	Lamiaceae	H	Wp	Abdominal pain, gas trouble	0.90	0.65	36
*Mentha royleana* Wall. ex Benth.	Safaid podina	Lamiaceae	H	L	Diarrhea, vomiting	0.69	0.44	24
*Oxalis corniculata* L.	Khati buti	Oxalidaceae	H	Wp	Vitamin C deficiency, mouth smell	0.86	0.56	31
*Plantago major* L.	Bartang	Plantaginaceae	H	Sd	Diarrhea, Obesity	0.90	0.47	26
*Polygonum plebeium* R.Br.	Rani pal	Polygonaceae	H	Wp	Cough	0.45	0.25	14
*Portulaca oleracea* L.	Lunak	Portulacaceae	H	L	Diuretic	0.50	0.31	17
*Punica granatum* L.	Daruna	Lythraceae	S	Fr	Gas troubles, indigestion	0.81	0.55	30
*Ricinus communis* L.	Arand	Euphorbiaceae	S	Sd	Constipation	0.57	0.29	16
*Salvia moorcroftiana* Wall. ex Benth.	Kaljari	Lamiaceae	H	L	Cough, diarrhea	0.37	0.20	11
*Silybum marianum* (L.) Gaertn.	Kandyara	Asteraceae	H	Sd	Liver problems	0.52	0.31	17
*Taraxacum officinale* F.H. Wigg.	Dhodal	Asteraceae	H	L	Diabetes	0.76	0.51	28
*Trachyspermum amii* (L.) Sprague	Ajwain	Apiaceae	H	Fr	Diuretic, kidney stone removal	0.66	0.38	21
*Trifolium repens* L.	Shaftal	Fabaceae	H	Wp	Fever	0.63	0.44	24
*Verbascum thapsus* L.	Gidar tambacu	Scrupholariaceae	H	L	Diarrhea	0.45	0.24	13
*Vitex negundo* L.	Marwani	Verbenaceae	S	L	Indigestion, stomachache, gas trouble	0.59	0.36	20
*Xanthium strumarium* L.	Katula	Asteraceae	S	Fr	Diuretic, kidney stone removal	0.29	0.20	11
*Zanthoxylum armatum* DC.	Timber	Rutaceae	S	Fr	Abdominal pain, indigestion	0.69	0.44	24

**Table 3 plants-08-00581-t003:** Traditional folk recipes of medicinal plant species used in the study area.

Botanical Name	Dosage	Traditional Folk Recipe
*Ajuga integrifolia* Buch.-Ham.	Drink twice a day (240 mL)	Fresh plant crushed and extract is taken. Mix 15 g powder in a glass of water.
*Bauhinia variegata* L.	Drink twice a day (240 mL)	Roots are shade dried and grinded. Mix 15 g powder in a glass of water.
*Cannabis sativa* L.	Drink once in a week (240 mL)	Fresh leaves are shade dried, and grinded powder (15 g) is mixed with almond and khaskhaas in milk. Mostly used in severe summer. This drink is locally named (Sardai).
*Dysphania ambrosioides* (L.) Mosyakin & Clemants	A cup of tea once in a day	Seeds are sun dried, boiled in water with sugar to make tea.
*Cichorium intybus* L.	Drink half cup before breakfast	Roots are dried and grinded. Mix 5 g of powder in 2 L of water and boiled for 2 h, placed for a night in open sky.
*Convolvulus arvensis* L.	Drink once in a day (240 mL)	Whole plant is sun dried and grinded. 15 g of powder is mixed in a glass of water or milk.
*Indigofera heterantha* Brandis	Drink twice a day (240 mL)	Root bark dried and powdered. 5 g of powder and sugar mixed with water to make drink.
*Justicia adhatoda* L.	Drink twice a day (240 mL)	Shade dried leaves and roots are grinded. The powdered material and sugar is used to make tea.
*Malva parviflora* L.	Drink twice a day (240 mL)	Leaves are shade dried and grinded. Powdered material and ginger powder is mixed in water and boiled with the addition of sugar to make tea.
*Medicago sativa* L.	Drink twice a day (240 mL)	Fresh leaves are boiled in water with ginger for half an hour to make tea. Sugar is added for taste.
*Mentha longifolia* (L.) L.	Drink twice a day (240 mL)	Shade dried leaves are used to make tea by adding sugar.
*Mentha royleana* Wall. ex Benth.	Drink twice a day (240 mL)	Leaves are shade dried and grinded. Powdered material is used to make tea by adding sugar.
*Oxalis corniculata* L.	Drink twice a day (240 mL)	Fresh leaves are boiled in water for 2 h and placed for cooling.
*Plantago major* L.	For diarrhea, drink twice a day (240 mL) and for obesity, drink before breakfast (240 mL)	Half boil a glass of milk with sugar. Then, add 30 g of seeds in it. This drink is for diarrhea. In water, 15 g of seeds are dipped and placed for whole night. Then, in early morning, it is mixed in a glass of water (clean). This drink is for reduction of obesity.
*Polygonum plebeium* R.Br.	Drink twice a day (240 mL)	Fresh plant is boiled in water for 2 h with the addition of sugar to make tea.
*Portulaca oleracea* L.	Drink twice a day (240 mL)	Fresh leaves are boiled in water for 1 h, sugar added to make tea.
*Punica granatum* L.	Drink after meal (240 mL)	Sundried fruit epicarp is grinded. Then, 3 g of powder is mixed with same amount of *Citrus medica* epicarp and 45 g of black salt. Put its 15 g to a glass of water for making a drink.
*Ricinus communis* L.	Drink once in a day (240 mL)	Oil is extracted from seeds and its 2 drops is mixed in a glass of milk.
*Salvia moorcroftiana* Wall. ex Benth.	Drink twice a day (240 mL)	Fresh leaves are sundried and grinded. Then, 15 g of powder and honey each is mixed in water.
*Silybum marianum* (L.) Gaertn.	Drink once in a day (240 mL)	Ripened seeds are sundried and powdered. The grinded material is mixed in water to make a drink.
*Taraxacum officinale* F.H. Wigg.	Drink once in a day (240 mL)	Fresh leaves are boiled in water till it becomes viscous. Then, add 5 mL of this liquid to a glass of fresh water for making a drink.
*Trachyspermum amii* (L.) Sprague	Drink twice a day (240 mL)	Fresh fruits are sundried and grinded. Then, 15 g of sugar and powder each is added to a glass of water.
*Trifolium repens* L.	Drink twice a day (240 mL)	Whole plant is boiled in water for an hour. Then, add 15 g of sugar and place it to cool.
*Verbascum thapsus* L.	Drink twice a day (240 mL)	Shade dried leaves are powdered. Then, 15 g of grinded material is added to a glass of water.
*Vitex negundo* L.	Drink once in a day (240 mL)	Fresh leaves are mashed, and extract is filtered. Then, 15 mL extract is added to a glass of water.
*Xanthium strumarium* L.	Drink twice a day (240 mL)	Fresh fruits are sundried and powdered. Then, 15 g of powder is mixed in a glass of water.
*Zanthoxylum armatum* DC.	Drink twice a day (240 mL)	Fresh fruits are sundried and powdered. Add 5 g in a glass of water.

g = grams, mL = milliliter.

**Table 4 plants-08-00581-t004:** Fidelity level of reported medicinal plants for various ailments.

**A) Plants with High Fidelity Level and Reported for One Ailment (FL = 100%)**
Diabetes
*Ajuga integrifolia*
*Taraxacum officinale*
Obesity
*Bauhinia variegata*
Fever
*Dysphania ambrosioides*
*Trifolium repens*
Typhoid fever
*Cichorium intybus*
Diuretic
*Indigofera heterantha*
*Portulaca oleracea*
Gas trouble
*Malva parviflora*
*Medicago sativa*
Cough
*Polygonum plebeium*
Constipation
*Ricinus communis*
Liver problem
*Silybum marianum*
Diarrhea
*Verbascum thapsus*
**B) Plants Fidelity Level Reported with Most Cited Medicinal Uses**
Warmness
*Cannabis sativa* (FL = 93.75%)
Diarrhea
*Plantago major* (FL = 92.31%)
Gas trouble
*Mentha longifolia* (FL = 88.89%)
Mouth smell
*Oxalis corniculata* (FL = 74.19%)
Vomiting
*Mentha royleana* (FL = 70.83%)
Dysentery
*Convululus arvensis* (FL = 68.42%)
Diuretic
*Trachyspermum amii* (FL = 66.67%)
*Xanthium strumarium* (FL = 63.64%)
Throat infection
*Justicia adhatoda* (FL = 61.11%)
Indigestion
*Zanthoxylum armatum* (FL = 62.50%)
*Punica granatum* (FL = 60%)
Cough
*Salvia moorcroftiana* (FL = 54.55%)
Gas trouble
*Vitex negundo* (FL = 45%)

**Table 5 plants-08-00581-t005:** Comparative analysis of the present study with previously published literature.

Botanical Name	Similar Uses from Literature	Dissimilar Uses from Literature	Literature Comparison
*Ajuga integrifolia* Buch.-Ham.	NR	Jaundice, hypertension ^5,12^, malarial fever ^5^, abdominal pain ^7^, tonic, dental, hepatic stimulant ^11^, throat sore, colic, purifying blood, epilepsy ^12^	5●, 7●, 11●, 12●
*Bauhinia variegata* L.	Fatness ^14^	Anthelmintic, tonic, astringent, skin diseases, ulcer, scrofula ^4^, fuel wood ^4,8^, fodder, vegetable, skin diseases, leprosy, ornamental ^8^, gastrointestinal problems ^18^	4●, 8●, 14●, 18●
*Cannabis sativa* L.	Cold drink (Tandai) for Warmness ^12^	Bowels, dyspepsia, gonorrhea, tonic ^3^, sedative ^3,5^, anodyne, anti-lice ^5^, appetite, abdominal swelling ^7^, narcotic ^3,5,17^, anti-diarrheal, stimulant ^17^, stomachache ^18^	3●, 5●, 7●, 12■, 17●, 18●
*Dysphania ambrosioides* (L.) Mosyakin & Clemants	NR	Piles, stomach problems ^5^	5●
*Cichorium intybus* L.	NR	Bile secretion, digestion, stomachache, tonic, carminative, cordial, obstructed, menstrual disorder, dropsy ^3^, fever ^12^	3●, 12■
*Convolvulus arvensis* L.	NR	Bowels ^2^, brain tonic, diuretic, sedative, anti-septic, skin diseases ^3,5^, stomach problems ^5^, constipation ^7^, skin problems ^12^, purgative ^2,3,5,12^, anthelmintic, galactagogue ^17^	2●, 3●, 5●, 7●, 12●, 17●,
*Indigofera heterantha* Brandis	NR	Hepatitis, whooping cough, blackening of hairs ^2^, fuelwood ^4^, scabies, stomach problems ^12^, stomach problems ^14^	2●, 4●, 12●, 14●
*Justicia adhatoda* L.	Throat infection ^3,4,14^, cough ^20^	NR	3■, 4■, 14■, 20■
*Malva parviflora* L.	NR	Cold, cough, sore throat, blocked nose ^17^	17●
*Medicago sativa* L.	NR	NR	
*Mentha longifolia* (L.) L.	Gas expeller ^2^, stomach problem ^6^, abdominal pain ^19^	Aromatic, stomachache, stimulant, mouthwash, vomiting ^5^	2■, 5●, 6■, 19■
*Mentha royleana* Wall. ex Benth.	Vomiting, diarrhea ^2, 12^	NR	2■, 12■
*Oxalis corniculata* L.	Mouth smell ^14^	Anti-septic, skin curing, stomach troubles, cleaning rusted vessel ^3,5^, digestion ^12^, food ^16^	3●, 5●, 12●, 14■, 16●
*Plantago major* L.	NR	Constipation, intestinal problems ^5^, infected hooves ^7^, bedsores, candidiasis ^12^	2●, 5●, 6●, 7●, 12●,
*Polygonum plebeium* R.Br.	Cough ^14^	Pneumonia, bowels ^5^	2●, 5●, 14■
*Portulaca oleracea* L.	NR	Liver ^12^, kidney diseases ^12,14^, abdominal pain, diabetes, asthma ^14^	12●, 14●
*Punica granatum* L.	NR	Dysentery ^7,8,13,18^, cough ^8,12,17^, fuel wood, tonic, anthelmintic ^8,12^, skin diseases ^13^, heart palpitation, blood pressure ^14^, food ^16^. cold, flu, nasal decongestant ^17^, diarrhoea ^18^	7●, 9●, 12●, 13●, 14●, 16●, 17●, 18●
*Ricinus communis* L.	Constipation ^1,5,7^	Purgative, lubricant ^3^, bowels ^12^, rheumatic joints, swellings, inflamed muscle, toothache ^13^, purgative, anti-flatulent, anti-inflammatory, analgesic ^17^, laxative, diarrhoea ^18^, leg sore, piles, edema ^20^	1■, 3■, 5■, 7■, 12●, 13●, 17●, 18●, 20●
*Salvia moorcroftiana* Wall. ex Benth.	Liver problems ^3^	Abscesses, dysentery, colic ^2^, wounds healing ^12^, white juicy part of stem as food ^16^	2●, 12●, 16●
*Silybum marianum* (L.) Gaertn.	Jaundice, hepatitis, cirrhosis ^1^	NR	1■, 3■
*Taraxacum officinale* F.H. Wigg.	NR	Hepatitis, appetite ^1^, urination, kidney disorders ^2^, tonic ^2,6^, diuretic, laxative, anti-inflammatory, digestive disorders ^6^	1●, 2●, 6●
*Trachyspermum amii* (L.) Sprague	NR	Appetite stimulant, galactagogue ^7^	7●
*Trifolium repens* L.	NR	Worms removal ^2^	2●
*Verbascum thapsus* L.	Diarrhea ^3,5,7,14^	Respiratory tract infection, ear infection, dry cough ^1^, aphrodisiac, skin problems, irritability ^2^	1●, 2●, 3■, 5■, 7■, 14■
*Vitex negundo* L.	Gas troubles ^14^	Aromatic, tonic, febrifuge, diuretic, anthelmintic, headache, fuel wood ^8^, chest pain ^13^, skin problems, child crying, snake bite ^20^	9●, 13●, 14■, 20●
*Xanthium strumarium* L.	NR	Skin problems ^5^, malarial fever ^5,14^, carminative ^14^	5●, 12●, 14●
*Zanthoxylum armatum* DC.	NR	Vomiting, indigestion ^7^, fuel wood, digestion, aromatic, condiment, gum diseases ^8^, carminative, pile ^8,13^, malarial fever, antipyretic ^12^, stomachache ^12,13^, fever, cholera ^13^	7●, 8●, 12●, 13●

NR = not reported, ■ = similar uses reported and ● = dissimilar uses reported. 1 = Jamal et al. [52]; 2 = Khan et al. [42]; 3 = Qureshi et al. [28]; 4 = Jan et al. [50]; 5 = Matin et al. [41]; 6 = Ume-Ummara et al. [53]; 7 = Abbasi et al. [51]; 8 = Ahmad et al. [46]; 9 = Ajaib and Khan [47]; 11= Tariq et al. [49]; 12 = Akhtar et al. [29]; 13 = Rashid et al. [23]; 14 = Ijaz et al. [5]; 16 = Ahmad and Pieroni [48]; 17 = Ullah et al. [31]; 18= Kichu et al. [13]; 20 = Kadir et al. [30].

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
