# Peer review of "Herbal Teas and Drinks: Folk Medicine of the Manoor Valley, Lesser Himalaya, Pakistan"

_plants, 2019, doi:10.3390/plants8120581_

Round 1

Reviewer 1 Report

The Authors of a manuscript submitted to Plants have considered the ethnomedicinal aspects of species found in Pakistan, in particularly in the Mansehra region. There is no doubt that such kind of data are really necessary for documentation of folk medicinal plants. However, the research has completely missed the scope of the journal. No relation to the fields of structural, functional and experimental botany can be found. The research is rather theoretical work. In addition, classification of compounds has been failed (L. 48-50).

The scientific soundness is rather weak, please check the statements which are common, e.g. L. 47-48; section Study area, particularly L. 80-81, Figures 2 and 3. In this case all plants should have been documented (Figure 4). It would be better to move tables provided in the supplementary materials to the main manuscript.

The Discussion section requires improving because it is based mainly on the data previously reported about plants which pharmacological activity has been already well known. Otherwise the title is completely inadequate.

108 Please check “Collected specimens were poisoned…” English language requires a serious improvement by a native speaker; it is advised to use “such as” instead of “viz.” ; “obesity” instead of “fatness”; sometimes the Authors use capital letters when it’s not necessary, e.g. “Bowels”, “Jaundice”,“Hepatitis”.

The legend for Table 1 should be provided.

Figure 9 and 10 are not necessary.

Author Response

RESPONSE TO REVIEWER 1 comments

Reviewer 1

Point 1: The Authors of a manuscript submitted to Plants have considered the ethnomedicinal aspects of species found in Pakistan, in particularly in the Mansehra region. There is no doubt that such kind of data are really necessary for documentation of folk medicinal plants. However, the research has completely missed the scope of the journal. No relation to the fields of structural, functional and experimental botany can be found. The research is rather theoretical work. In addition, classification of compounds has been failed (L. 48-50).

Response to point 1: Dear reviewer, as you said our study is almost entirely theoretical. However, contrary to what has been stated, we followed both the aim and scope of the journal. According to the journal: “The main aim of our journal is to encourage scientists and research groups to publish theoretical and experimental results of research in all fundamental and applied fields of plant science.”

Moreover, the scope of the journal is very clear in relation to the type of study accepted: “Journal covers the following areas and sub-areas in plant science ... biology of medicinal plants; ethnobotany; biological effects of active substances from plants; phytomedicine”. Similar type of studies published in this esteemed journal are;

https://www.mdpi.com/2223-7747/7/4/104

https://www.mdpi.com/2223-7747/8/8/256

https://www.mdpi.com/2223-7747/8/8/246

Furthermore, this revised manuscript (Plants-635793) has been submitted to the special issue (Medicinal Plants) which completely covers the scope of the Journal’s special issue.

Point 2: The scientific soundness is rather weak, please check the statements which are common, e.g. L. 47-48; section Study area, particularly L. 80-81, Figures 2 and 3.

Response to point 2: We followed all recommendations and improved the manuscript. Line 48-50, the article (Audu et al. Phytochemical, proximate composition, amino acid profile and characterisation of Marijuana (Canabia sativa). J. Phytopharm. 2014, 3, 35–43). has been re-reviewed, and the statement has been corrected (See line 52-53). Figures 2 and 3 have been deleted as per suggestion.

Point 3: In this case all plants should have been documented (Figure 4).

Response to point 3: Dear reviewer, we have documented a total of 27 plant species and providing photos for all of them would make the figure and manuscript file size too much bulky, and the manuscript size will be increased which may exceed the journal’s upload file limit, but if it’s necessary then will provide that.

Point 4: It would be better to move tables provided in the supplementary materials to the main manuscript.

 Response to point 4: Dear reviewer, as per your suggestion Table S2 has been shifted to the main body of the manuscript (Table 4)

Point 5: The Discussion section requires improving because it is based mainly on the data previously reported about plants which pharmacological activity has been already well known. Otherwise the title is completely inadequate.

 Response to point 5: We improved the discussion section according to your fruitful suggestion.

Point 6: 108 Please check “Collected specimens were poisoned…”

 Response to point 6: Dear reviewer, the sentence is right. Poisoning the plant is a herbarium technique to prevent the plant specimens from fungal and pests attack. However, we improved the sentence.

Point 7: English language requires a serious improvement by a native speaker;

 Response to point 7: We improved the English of the manuscript

Point 8: it is advised to use “such as” instead of “viz.” ; “obesity” instead of “fatness”;

 Response to point 8: As per reviewer’s suggestion, all changes were made accordingly.

Point 9: sometimes the Authors use capital letters when it’s not necessary, e.g. “Bowels”, “Jaundice”, “Hepatitis”.

 Response to point 9:We have corrected that, rechecked and improved the manuscript for other typos as well.

Point 10: The legend for Table 1 should be provided.

 Response to point 10: The legend of Table has been provided.

Point 11: Figure 9 and 10 are not necessary.

Response to point 11: We removed both the figures as per your suggestions

Reviewer 2 Report

  I think it's a great job, especially because of the methodology, which can serve as an example and demonstration of other work that can be done in other parts of the world by young teams that are starting in quantitative Ethnobotany. The fact of incorporating this quantitative character to the studies of wild food and medicinal plants is very important to raise the scientific quality of our Ethnobotanical publications.
Another interesting methodological aspect is to locate research in areas of difficult access, where traditional knowledge is better preserved.
The results are adequate, correct, and the discussion and conclusions do not exaggerate the evidence of what was found. The sample sizes are reduced due to the difficulties presented in the Material and Method. In this regard, I would like to invite the authors to break down the knowledge of men and women in future works, with the hypothesis that it may be very different. In the sample of this studied population, I perceive a low female percentage.
I congratulate the authors for the idea of ​​incorporating the indices of similarity and diversity classically used in Plant Ecology for these studies. I find it magnificent.
Congratulations to the authors, it seemed like a great job

Author Response

Reviewer 2

 I think it's a great job, especially because of the methodology, which can serve as an example and demonstration of other work that can be done in other parts of the world by young teams that are starting in quantitative Ethnobotany. The fact of incorporating this quantitative character to the studies of wild food and medicinal plants is very important to raise the scientific quality of our Ethnobotanical publications.
Another interesting methodological aspect is to locate research in areas of difficult access, where traditional knowledge is better preserved.
The results are adequate, correct, and the discussion and conclusions do not exaggerate the evidence of what was found. The sample sizes are reduced due to the difficulties presented in the Material and Method. In this regard, I would like to invite the authors to break down the knowledge of men and women in future works, with the hypothesis that it may be very different. In the sample of this studied population, I perceive a low female percentage.
I congratulate the authors for the idea of ​​incorporating the indices of similarity and diversity classically used in Plant Ecology for these studies. I find it magnificent.
Congratulations to the authors, it seemed like a great job 

Dear reviewer, we are very delighted and feel glad to see your response on our manuscript which completely fits on your standard. Thanks for such nice appreciation and for your fruitful suggestions as well. In further studies, we will take the gender in account.

Reviewer 3 Report

Herbal Teas and Drinks: Folk Medicine of the Manoor Valley, Lesser Himalaya, Pakistan

 I write you in regards to manuscript # plants-635793 entitled "Herbal Teas and Drinks: Folk Medicine of the Manoor Valley, Lesser Himalaya, Pakistan" which you submitted to the Plants.

Authors need to follow the following instructions to improve this manuscript

Page 1, Line 17-18: Check the author affiliation serial number Page 1, Line 17-18: Author should bring an important point in the Abstract Page 2: In Material and Methods: The author should mention the tea preparation or infusion method/technique/process. For more information please check: https://onlinelibrary.wiley.com/doi/10.1002/cbdv.201900459 Page 5, Figure 6: Author should add standard deviation or standard error in Figure 6 Page 6-8, Table 1: In literature comparison (It is mandatory to add a footnote for the symbols) Page 11-13, Table 2: The author should mention the exact volume of tea/drinks/extract. 1 glass or 1 tablespoon or 1 cup is not specific. Just mention in a milliliter. Page 13, Line 178: Author added reference in result part. I think it will go into the discussion. Discussion: should rewrite carefully with relevant references. Present discussion as like as the second version of Introduction. Why mentioned plants are important for human health? The author should gather that information especially bioactive compounds and antioxidant capacity or medicinal properties of the mentioned plants. Specifically, should mention the function of a specific plant. Relevant references should enrich. References: should follow the journal guideline Please check carefully before resubmission. English correction is needed from a native speaker or proofreads company (grammar, spell, writing sequence, etc.)

I recommend to improve the manuscript and resubmit.

Author Response

Reviewer 3

I write you in regards to manuscript # plants-635793 entitled "Herbal Teas and Drinks: Folk Medicine of the Manoor Valley, Lesser Himalaya, Pakistan" which you submitted to the Plants.

Authors need to follow the following instructions to improve this manuscript

Point 1: Page 1, Line 17-18: Check the author affiliation serial number

Response to point 1: All affiliation numbers were checked and corrected.

Point 2: Page 1, Line 17-18: Author should bring important point in the Abstract

Response to point 2: Dear reviewer, we provided the main results of our study. Unfortunately, the journal does not provide much room to detail our findings. Therefore, we prioritized the main ones.

Point 3: Page 2: In Material and Methods: Author should mention the tea preparation or infusion method/technique/process. For more information please check: https://onlinelibrary.wiley.com/doi/10.1002/cbdv.201900459

Response to point 3: As per your suggestion, the tea and drink preparation method has been mention (see section 2.3 Tea and drink infusion preparation)

Point 4: Page 5, Figure 6: Author should add standard deviation or standard error in Figure 6

Response to point 4: Dear reviewer, this graph shows absolute numbers (not replicates), disregarding the need of errors.

Point 5: Page 6-8, Table 1: In literature comparison (It is mandatory to add foot note for the symbols)

Response to point 5: Symbols have been added (see foot note of Table 4)

Point 6: Page 11-13, Table 2: Author should mention the exact volume of tea/drinks/extract. 1 glass or 1 table spoon or 1 cup is not specific. Just mention in milliliter.

Response to point 6: Dear reviewer, following your valuable suggestion, the amount with units has been added.

Point 7: Page 13, Line 178: Author added reference in result part. I think it will go in discussion.

Response to point 7: We removed the reference and adjusted the text, that was mistakenly placed.

Point 8: Discussion: should rewrite carefully with relevant references. Present discussion as like as second version of Introduction. Why mentioned plants are important for human health? Author should gather those information especially bioactive compounds and antioxidant capacity or medicinal properties of the mentioned plants. Specifically, should mention the function of specific plant.

Response to point 8: We improved our discussion and removed the lines related to introduction following reviewer’s suggestions.

Point 9: Relevant reference should enrich.

Response to point 9: We provide the main references of the topic, especially the references related to the area of study.

Point 10: References: should follow the journal guideline. Please check carefully before resubmission.

Response to point 10: We rechecked the guidelines and formatted all of them as per journal’s guidelines.

Point 11: English correction is needed from a native speaker or proof read company (grammar, spell, writing sequence, etc.)

Response to point 11: We improved the English of the manuscript

Round 2

Reviewer 1 Report

Firstly, the mansucript has been difficult to read in the present form with all editing comments. Please move Figure 2 to the Supplementary materials.

Author Response

Response to Reviewer 1 Comments

Point 1: Please move Figure 2 to the Supplementary materials.

Response to point 1: Dear reviewer, as per your suggestion, Figure 2 has be moved to Supplementary data file (Figure S1). The attached manuscript is now in revised clean form, not in tracking version. Whereas, in the online system tracking version in pdf has also been attached.

Reviewer 3 Report

Comment: 1) Figure 3 should change to Table.

Author Response

Response to Reviewer 3

Point 1: Figure 3 should change to Table.

Response to Point 1: As per your suggestion, figure 3 has been replaced with table 1. Moreover, necessary modifications have been done in the manuscript (see attached tracking file)
